# Influence of Pore Networking and Electric Current Density on the Crack Pattern in Reinforced Concrete Test Due to Pressure Rust Layer at Early Ages of an Accelerated Corrosion Test

**DOI:** 10.3390/ma12152477

**Published:** 2019-08-04

**Authors:** Ángela M. Bazán, Encarnación Reyes, Jaime C. Gálvez

**Affiliations:** Departamento de Ingeniería Civil, Construcción, E.T.S de Ingenieros de Caminos, Canales y Puertos, Universidad Politécnica de Madrid. C/Profesor Aranguren 3, s/n, 28040 Madrid, Spain

**Keywords:** corrosion, concrete cover, cracking, SEM image analysis, rust layer, strain gauge, pore size

## Abstract

Research on early stages of corrosion of steel bars caused by chloride penetration is relevant in improving the durability of reinforced concrete structures. Similarly, the formation and development of cracks induced in the surrounding concrete is also of great importance. This paper uses integration of the analytical models examined in the published literature, combined with experimental research in corrosion induced at the concrete/steel interface, in estimating the time-to-crack initiation of reinforced concrete subjected to corrosion. This work studies the influence of the porous network and electric current density on the cracking process at early ages. The experimental program was performed by using an accelerated corrosion test. Two types of concrete were performed: A conventional concrete (CC) and a concrete with silica fume (SFC). A current density of 50 μA/cm^2^ and 100 μA/cm^2^ was applied to specimens of both concretes. Examination performed by scanning electron microscopy (SEM) and energy-dispersive X-ray spectroscopy (EDS) provided both qualitative and quantitative information on the penetration of the rust layer in the surrounding concrete porous network. Strain gauges were used to measure corrosion-induced deformations between steel and concrete matrices, as well as the formation of corrosion-induced cracks. A good correlation between the rate of penetration of the rust products in the surrounding pores and the delay of the cracking pressure in concrete was observed from the experimental results. This phenomenon is incorporated into the analytical model by using a reduction factor, which mainly depends on the pore size of the concrete. The crack width obtained exhibited a significant dependency on electric current density at the beginning of the test, depending mainly on the pore size of the concrete later.

## 1. Introduction

Durability of reinforced concrete is a significant research line, given the high costs associated with the maintenance and repair of structures. Corrosion of steel-rebar reinforcement is one of the most serious and frequent deterioration mechanisms identified in reinforced concrete structures. This deterioration process frequently causes significant structural problems that eventually lead to collapse [1,2]. Steel embedded in normal reinforced concrete is protected from corrosion due to a passive film, formed on the surface of the steel because of the high alkalinity of pore solution in the concrete. The break of the passive barrier would occur with the pH value being lower than 10 or, most commonly, by the presence of chloride ion.

Once the passive film is broken, corrosion starts, and rust products are produced by an electrochemical reaction of the iron. The rust produced occupies a larger volume than the original material. At an early stage, this extra volume can be accommodated by a region of the porous network of the concrete that surrounds the rebar, delaying the tensile stress development in the concrete/steel interface [3,4]. Therefore, this region of concrete, called the corrosion accommodation region (CAR), has a major influence on the prediction of the crack initiation time and crack propagation [4,5,6].

The CAR thickness depends on several factors, in particular the volume and connectivity of the pores in the concrete/steel interface and the corrosion current density. The microstructural characteristics of the concrete/steel interface, in turn, depend on the surface area of reinforcement, w/c ratio, degree of hydration, degree of consolidation and, among others, type of cement. Thus, accurate prediction of the CAR size and crack initiation time is highly complex. Several published authors have proposed a fixed value for the CAR thickness, which due to expansive pressure must be filled with corrosion products before tension stress development [5,7]. Recent works have examined the formation and propagation of corrosion-induced cracks and the CAR thickness by using the X-ray attenuation technique [4,8]. 

Once the CAR is filled, the corrosion products cause expansive pressure on the surrounding concrete, with tensile stresses emerging on the concrete/steel interface [5]. Eventually, this can lead to cracking in the case of the tensile stresses exceeding the tensile strength of the concrete [9].

The formation of cracks provides a path through which the aggressive agents from the environment can easily penetrate and reach the reinforcement, accelerating the corrosion process. In this stage, the corrosion products begin to fill the cracks, which then cause crack opening and propagation, spalling and delamination of the cover [3,10]. Figure 1 shows the three main stages of the corrosion process.

To improve knowledge about durability of reinforced structures in chloride environments is necessary to obtain a thorough understanding of the corrosion process from its beginning. Particularly important is a detailed study of the CAR formation in order to predict time-to-crack initiation, the propagation period and ultimate service life. Many of the experimental studies on corrosion-induced crack initiation and propagation conducted previously have used high corrosion rates to accelerate the process on concrete with embedded rebar specimens being monitored with strain gauges on the surface [11,12]. Techniques based on the use of microscopy have been used to characterise the concrete/steel interface [13]. The authors of this paper propose combining both experimental methods by using strain gauges placed as closely as possible to the rebar to assess the radial displacement of the surrounding concrete and scanning electron microscopy (SEM), along with energy-dispersive X-ray spectroscopy (EDS) to examine the concrete/steel interface. This combined method exhibited an adequate performance to study the corrosion process from the earliest stages in a previous published work [14]. In this paper, however, results of an experimental work conducted to investigate the influence of the pore size and corrosion current density on the CAR and cracking process by using the proposed procedure are presented. The experimental program used accelerated corrosion tests [15,16,17] with current densities of 50 μA/cm^2^ and 100 μA/cm^2^, usually used for accelerated corrosion studies [4,12,18,19,20]. Two concretes mixtures were performed: One without admixtures (CC) and another with silica fume (SF). The objective of using SF in one mixture was to obtain two concretes with distinct microstructures in order to study the influence of the porous network on the CAR as well as the formation and propagation of corrosion-induced cracks. The specimens used had a steel rebar embedded in the middle. The strain of the concrete due to corrosion of the rebar was monitored by means of strain gauges. The aforementioned SEM and EDS were used to examine the penetration of the rust corrosion products found in the concrete. Lastly, the paper uses an analytical model based on the thick-walled cylinder approach [10] to reproduce the experimental results. This approach has been widely used in previous studies described for instance in [21]. In this work, the crack width is predicted by using a combination of wavelength dispersive spectroscopy along an analytical line that covers the interface and the radial displacement assessed by the gauges. Furthermore, the crack initiation time is estimated from converting the retrieved strain data into stresses and by comparing them with the tensile strength.

## 2. Experimental Program

### 2.1. Materials, Mix Proportioning and Specimens

Two concrete mixtures were designed trying to obtain similar properties and approximately the same percentage of total porosity though different pore microstructure. As a result, two concretes with quite similar overall porosity, but different pore size distributions were obtained. Portland cement CEM I 52.5 R and Portland cement CEM I 42.5 R, in accordance with the standard EN 197-1:2011, were used for the preparation of the two concrete mixtures. The first cement type was used to elaborate a conventional concrete (CC) without admixtures and the second one to elaborate a silica fume concrete mixture (SFC) by using SF, with 91% of silica dioxide in its composition, as mineral admixture. The second mixture uses SF with 10% of replacement percentage weight of cement and two as an efficiency factor [22]. The water/binder ratio was 0.45 in both cases. Chlorides were added to the mix in order to generate depassivation of the steel [6,23]. To this end, 3% of CaCl_2_ by weight of cement was added to each concrete [6,23,24]. Furthermore, a polycarboxylate-based superplasticizer (SP), named Sika ViscoCrete 5720, was added to improve workability. Mix proportioning is shown in Table 1.

In order to study the evolution of the corrosion products at different ages from early stages and with two current densities, 16 specimens of reinforced concrete were cast for each concrete type with the following dimensions: 15 × 15 × 15 cm^3^. Rebars of diameter of 20 mm were treated with the aim of protecting the steel from corrosion initiators, propagating points and preferential attacks. First, rebars were coated entirely with epoxy resin. Later, the rebars were machined and the diameter reduced by up to 12 mm along 100 mm, beginning at a point located 30 mm from the extreme of the specimen, as shown in Figure 2.

With the objective of monitoring the tangential displacement of the surrounding concrete induced by the corrosion, strain gauges were embedded in three specimens per concrete type and current density. For such a purpose, these specimens were cast in two times, casting one half of the volumes in each of them. When the first half was set, four strain gauges were placed perpendicularly to each other at the concrete surface around the steel rebar as closely as possible [25]. A resin appropriate for monitoring purposes was used to glue the strain gauges. Lastly, a single component solvent-based polyurethane lacquer, PU140, was used as protective film on the gauge, in order to avoid any potential problem derived from the elevation of the humidity caused by the casting of the second half. Figure 3 shows the strain gauges around the rebar. Table 2 shows the characteristics of the strain gauges.

### 2.2. Mechanical Test

Four experimental tests were carried out in order to study the mechanical properties of the concrete. The compressive strength, the elastic modulus and the tensile strength were measured at 28 days on cylindrical φ150 mm × 300 mm specimens. Compression strength tests were performed according to EN 12390-3. Three specimens were tested for each type of concrete. The elastic modulus was estimated in accordance with standard EN 12390-13. The indirect tensile strength was obtained according to standard EN 12390-6 by a tensile splitting test on three specimens.

Lastly, fracture energy was assessed according to the recommendation RILEM TC-187 SOC. Three specimens of each concrete type were tested under three-point bending testing procedure. The dimensions of the specimens were 100 × 100 × 430 mm^3^ with a notch at the centre of 1/3 of the height of the beam. The mechanical properties of the two concretes studied are showed in Table 3.

### 2.3. Accelerated Corrosion Tests and Test Program

The experimental set-up used is shown in Figure 4a. The concrete-rebar electric circuit was externally closed. A constant anodic current was applied to enforce corrosion. The electric circuit joined the steel rebar, the anode, with a lead sheet being placed on the surface of the slabs which acted as a cathode. The electrical connection was performed by partially immersing the specimens in tap water. The water level was maintained at approximately 1cm height. In order to induce corrosion, 3% of CaCl_2_ by weight of cement was added to the mix [19]. 

In this study, specimens were subjected to current densities of 50 and 100 µA/cm^2^ in order to examine the influence of velocity of corrosion in the CAR formation. These values allow a sufficiently short experiment but keep induced corrosion within values which might be found in practice [19]. With such an objective, the intensity density must stay somewhat low (<200 μA/cm^2^) with respect to Faraday’s law, otherwise a significant increase of in the tension–strain response and, consequently, in the cracking process might occur. Theoretically, intensity densities could reach 500μA/cm^2^, but non-negligible variations might appear, and the accuracy of the experiment could not be guaranteed [20].

Two independent circuits were assembled: The first one comprised only the specimen with strain gauges, and the second one the rest of the specimens by using a parallel system. Specimens from the second circuit were disconnected and extracted at different ages in order to provide study by means of SEM. Since a constant current density of 50 and 100 μA/cm^2^ was imposed, the total current in each circuit could be computed. Results of the surface rebar electricity density and total electricity current in each circuit are shown in Table 4.

### 2.4. Microscope Analysis

The study of the development of the corrosion process and progress of the rust corrosion layer (CL) were observed and analysed by means of SEM and EDS (JEOL Superprobe JXA-8900 M, Tokyo, Japan). The specimens were disconnected and cut at the following ages: One, two, five, eight, 14, 21, 26 and 35 days. First, prismatic samples containing the rebar with a square base of 50 × 50 mm^2^ were cut. Then, thin slices of approximately 10 mm thickness were obtained for SEM observation. All the cuts were made with the precision diamond cut-off machine STRUERS SECOTOM-10, with a 0.8 mm thick disc running across the prismatic sample. In order to avoid a washing of the corrosion products, liquid petroleum jelly was used to cool the cutting disc. Subsequently, these slices were prepared for SEM observation (see Figure 4b). To this end, the samples were inserted in a vacuum chamber until constant weight and then impregnated with a low-viscosity epoxy resin. The resin was hardened for 48 hours. At a later stage, all the samples were polished by using a lapping and polishing machine with silicon carbide abrasive papers from grades 240 to 1200 lubricated with water. Polishing was performed by using Wenol® metal polish to improve imaging quality. Lastly, a sputter conductive coating of carbon was applied to the polished area in order to obtain a higher resolution image, allowing a more detailed examination. 

A field-emission scanning electron microscope (JEOL Superprobe JXA-8900 M, Tokyo, Japan) and operated in backscattered electron (BSE) mode was used to create the element maps of the steel/concrete interface at different ages. On this basis the time-dependent corrosion product development, the CAR and subsequent formation and propagation of cracks were studied. Four chemical elements were analysed to examine rust development on the concrete/steel interface: Calcium, iron, oxygen and silicon. The results are represented in one general image. Different colour intensities, ranging from black over green to purple, make observation of layers in each element map possible. Steel rebars and the corrosion layer are brighter in the images due to the oxygen content. In the case of concrete, Ca and Si are found. An analytical line was selected across the steel/concrete interface on each image. The size of the reference region of each image depended on the age. Boundary conditions are shown in Table 5.

### 2.5. Pore Size 

Figure 5 shows frequency distribution of the pores for the mixtures studied and the distribution of macro and micropores for the SFC and the concrete without any additions (CC) at 28 days of curing. In this work, macropores are considered as those with a diameter greater than 50 nm in width according to [26]. Table 6 shows the percentage of total porosity, the average pore diameter and the median pore diameter for each concrete.

This work sought the influence of the pore size by maintaining the rest of the properties relatively constant. As Table 6 shows, the total porosity is quite similar in each concrete. Nevertheless, the average pore diameter and the median pore diameter are significantly higher in CC in comparison with SFC. Figure 5 shows the distribution of macro and micropores in each concrete, with a significantly higher amount of micropores being observed in SFC in comparison with the conventional concrete. Therefore, the concrete with silica fume has a denser network and a smaller average pore diameter, while presenting a similar overall porosity. The median pore diameter of SFC is half that of the pore diameter of CC. The general smaller diameter of the pores in SFC in comparison with CC could have a notable influence on a less accommodating capacity of the rust products.

## 3. Results

### 3.1. Experimental Displacement Field Measurements at Steel/Concrete Interface with Strain Gauges

The strain gauges were placed as closely as possible to the steel rebar in order to measure the strain as the rust products were growing. The data were checked daily and recorded. Figure 6 shows the circumferential strain around the rebar versus time for each concrete type and density. The value corresponds to the average circumferential strain measured by the embedded gauges in two specimens for each case studied. The coefficient of variation of the measured circumferential strain among specimens in each type of concrete and current of electric density was lower than 15%.

Based on the circumferential strain shown in Figure 6, the time of significant increment of the tension stress initiation on the concrete/steel interface may be estimated. The results obtained for 100 μA/cm^2^ show a linear pattern from the very beginning. In the case of the radial stresses with 50 μA/cm^2^, a bilinear pattern may be approached. Therefore, in this case, the strain gauges did not measure strain over a period of time, corresponding with the formation of the CAR. The time needed to accommodate the rust products (the horizontal part) is quite similar, around 120 h, in each specimen. Once the CAR was completely filled with rust products, expansive pressure started with a linear increment. During the formation of cracks, it is worth noting than the circumferential strain measured by the gauges might have an additional component due to micro-cracking before the formation of the macroscopic cracking. This process is likely to occur together with a possible injection of corrosion gel into the micro-cracks during the process. After the initiation of cracking, part of the circumferential stress is released.

### 3.2. Microstructure Characteristics

Mapping was adopted to observe the development of corrosion products at the steel/concrete interface. The areas with high levels of Fe are shown in purple and the rust layer represented in green. In order to complement the study, the distributions of corrosion products were analysed by EDS/WDS along an analytical line. Four chemical elements were chosen for this analysis: Fe, O, Si and Ca. Depending on the availability of these elements, the three main layers involved in the corrosion process could be distinguished: Steel, mill scale (CL + CAR) and concrete. As an example, Figure 7 shows the amount of Fe at two, five and eight days for the CC mixture under a current density of 100μA/cm^2^ by using the colour code. In the graphs below, the distributions of Fe and O are shown. The horizontal axis represents the distance from the starting point of the analytical line and the vertical axis the counts of photoelectrons per second.

The starting point of the analytical line and the vertical axis represent the counts of photoelectrons per second.

In the image corresponding to 35 days, the end of the filling of the ΔR can be distinguished. This specimen had a rust layer (CL) of 18 μm and a layer where corrosion products were diffusely distributed (CAR). Some authors refer to this as the ‘corrosion filled paste’ [27,28] when describing a two-step process in the corrosion distribution. This indicates that a corrosion layer is first formed around the steel rebar which then penetrates through the concrete. Based on these results, the mill scale (CL + CAR) thickness of the rust products can be determined in all the cases at 100 µA/cm^2^ and 50 µA/cm^2^. As Figure 8 shows, the current intensity and the type of concrete had a considerable effect on the mill scale thickness at the same age (14 days).

Table 7 shows the evolution of the corrosion in time, where the values of the mill scale thickness (ɅR) as a function of time are collected. On the basis of these values, it could be concluded that for the first 5 days no significant increase in mill scale thickness was detected in the specimens under a current density of 50µA/cm^2^. Afterwards, a linear growth could be appreciated.

Although examination of the concretes studied has not shown significant change in total porosity, a considerable pore refinement has been obtained by adding silica fume. SFC presented a reduction of 15.2% in the percentage of macropores, increasing the percentage of micropores with regard to CC. This might explain the differences obtained in the corrosion layer development, even though the variation exhibited between different concretes and intensities was relatively small.

## 4. Discussion

In order to compare the behaviour of the four cases studied with two current densities and two concrete types, the difference in current density used in the test of the circumferential strain development had to be eliminated. This could be achieved by multiplying the time by intensity to obtain the total electric charge. The intensity is obtained by multiplying the current density by the surface of the rebar subjected to corrosion, that is, the same in all cases and equal to 37.7 cm^2^ (see Figure 2). The specimens subjected to a current density of 100 μA/cm^2^ corresponded to a total current of 3.77 mA applied. The specimens under 50 μA/cm^2^ were subjected to a total current of 1.89 mA. Figure 9 shows the circumferential strain measured by the gauges in the vertical axis versus the total electric charge applied to each concrete in the horizontal one. As can be observed, curves corresponding to the lowest current density exhibited a bilinear pattern. Both concrete types needed around 500 C, and thus a similar time range, to induce notable pressure and therefore circumferential strain. After this turning point, at around 500 C, each concrete showed a linear pattern. The regression line for this part is the same for both concrete mixtures, with a quite good R^2^ coefficient. In the case of the highest current density, each concrete exhibited a linear pattern of the circumferential strain, and thus the circumferential stress, from the beginning. In this case, the regression line is similar for each concrete mixture and the slope slightly lower than the regression line corresponding to SFC50 and CC50.

The regression coefficient R^2^ obtained for SFC100 is considerably worse than that obtained in other cases. This might have been caused by the breaking of three of the four gauges used in the measurement of the circumferential strain in this case, as shown in Figure 10. 

The results displayed by Figure 9 and Figure 10 show that the placement of the four gauges around the steel bar might offer an improvement for cracking models versus those where they are placed on the concrete surface. The data acquired are in general a good representation for the development of stress throughout the corrosion process.

Figure 11 shows the experimental circumferential strain as a function of the rust layer thickness during the corrosion process and the regression line obtained. As can be seen, all cases presented a linear regression with a good R^2^ coefficient. The slope of the regression line was the same for each concrete at different current intensities and only slightly lower for CC mixtures in comparison with SFC mixtures. The rapid development of the corrosion process in SFC100 mixture, together with the smaller average pore diameter size in comparison with the conventional mixture, causes a lower capability to accommodate the corrosion products, producing larger circumferential deformations than other cases from the beginning.

## 5. Analytical Verification

Much published research has offered analytical models that describe the cracking process based on a thick-walled cylinder of one or two layers in line with the theory of plasticity proposed by Timoshenko and Goodier [29]. The most commonly used model today is described by Den Ujil and Bigaj [30], who proposed a model derived from Tepfers’ theory to describe the relationship between radial displacement and radial compression stress at the interface steel/concrete (Equation (1)) and circumferential hoop strains based on the confining capacity of the material surrounding the rebar. 

(1)$$\frac{\Delta R}{r}+\;\frac{n\;w}{2\pi \;r}=\;\epsilon$$
where *w* is the width of crack, *r* is the radius, ΔR is the radial displacement, *n* is the number of cracks and ε is the circumferential strain, parameter that in this case includes the crack width.

As showed in Figure 1, three stages can be distinguished in the corrosion process. The second one starts when the CAR is completely filled with rust corrosion products, which causes stress initiation. Subsequently, it is possible to identify a third stage when the tensile stress equals the tensile strength causing the cracking initiation and subsequent crack development as rust products fill them. Recently, some authors have proposed a two-stage model [31]. In line with such an approach, after the initiation of the steel corrosion, the penetration of rust products into the porous microstructure occurs together with the formation of the corrosion layer at the interface steel/concrete. Following this model, the CAR and CL have been assessed jointly in this work.

The procedure proposed combines use of strain gauges and microscopic techniques in order to obtain information about the circumferential hoop strains produced due to the corrosion process. The experimental tests results allow the estimation of the circumferential strain and the radial displacements. The first value is obtained through the measurements of the strain gauges. In the second case, the value is estimated from the development of the rust layer shown by the wavelength dispersive X-ray spectroscopy (WDS) curves. 

Once the corrosion process has started, it is possible to differentiate three radii: *r_0_* is the initial radius of the rebar; *r_1_* is the radius of the non-corroded rebar; and *r_2_* is the total radius of the corroded rebar, value that includes CL and CAR. Figure 12 shows a sketch of these radii, according to which Equation (2) is obtained: (2)$$r_{2}=r_{0}+\Delta R$$

Furthermore, Equation (3) establishes the equivalence of CAR + CL in this approach: (3)$$\;\mathrm{CAR}+\mathrm{CL}=r_{2}-r_{1}=\Delta R^{*}$$
and $\Delta R^{*}$ corresponds to the green area shown in Figure 7 and Figure 8.

With the aim of comparing the experimental circumferential strain provided by the experimental strain gauges measurements, placed as closely as possible to the rebar with the analytical approach, it is necessary to consider the geometrical dimensions of the strain gauges. Therefore, another radius is defined by adding the radius of the rebar with the half of the width of a strain gauge (Equation (4)), due to it being in contact to the surface of the steel [14]. 

(4)$$r^{*}=r_{0}+\frac{1}{2}e$$
with *r** being the radius of the strain gauge measurement (sum of the radius of the rebar with the half of the width of the strain gauge [14]) and *e* the breadth of the strain gauge.

It should be noted that in order to evaluate the circumferential hoop strain, the entire corrosion layer thickness indicated in the WDS images, $\Delta R^{*}$, cannot be directly used, since it includes the CAR. In this work, a reduction factor *f* is proposed [14] with the aim of relating $\Delta R^{*}$ with the effective radial displacement ($\Delta R_{effective})$ provided by the strain gauges measurements. Thus, the effective radial displacement is obtained by adding a reduction factor *f* in the elastic regime (*n* = 0). 

(5)$$\Delta R_{\mathrm{effective}}=\;\Delta R^{*}f$$

Subsequently, the circumferential strain (Equation (1)), which before the opening of the crack or cracks is the measurement assessed by the strain gauges (*ε*_sg_), can be expressed as: (6)$$\frac{\left(\Delta R^{*}f\right)}{r^{*}}+\frac{\omega n}{2\pi r^{*}}=\varepsilon \;\;\;\;\mathrm{with}\;\;\omega =0\;\overset{yields}{\to }\;\frac{\left(\Delta R^{*}f\right)}{r^{*}}+\frac{0n}{2\pi r^{*}}=\varepsilon =\varepsilon_{sg}$$
where $r^{*}$ is the radius of the strain gauge measurement (sum of the radius of the rebar with the half of the width of the strain gauge [14]), $\Delta R^{*}$ is CL + CAR, *f* is the reduction factor, *w* is the width of crack, *n* is the number of cracks, *ε* is the circumferential strain that includes the opening of the cracks. 

The reduction factor can be obtained from Equation (6). Table 8 and Table 9 show the reduction factor calculated for each concrete mixture at each age studied. The average reduction factor is 0.005 for CC, which is slightly higher than the value obtained for SFC, 0.006. Therefore, it could be said that a pore refinement might influence in the decrease of the reduction factor, as in the case of SFC. By contrast, the current intensity does not appear to have significant influence in the reduction factor value in the cases studied.

With a view to study the evolution of the crack width with time once the cracks initiate, the average reduction factor obtained from Equation (7) can be used at the ages studied. Table 10 shows the crack widths in all cases according to Equation (7). Figure A1 shows the cracks surrounding the rebars of the specimens at several ages of testing. At the moment of crack initiation, some of the strain gauges were broken and crossed by a crack trajectory, though the rest of strain gauges continued measuring circumferential strain. As shown in Figure 3, there were four strain gauges that covered the potential directions of the radial cracks, except those that crossed the contact between each pair of strain gauges. In such a way, the detection of the radial crack initiation could be assured due to the breakage of any of the strain gauges. Additionally, a slight fall in the strain measured by the unbroken strain gauges is observed at the moment of cracking (see Figure 10). This slight fall in the strain measured is produced when cracking starts, probably due to a reduction in stiffness of the thick cylinder under radial pressure by the rust layer. Part of the circumferential stress is released after the initiation of crack, and some more expansion is required before reaching again the radial pressure. 

After completion of the test, prisms of 28 × 28 × 5 mm^3^ containing the rebar at the centre were cut with the aim of observing the number and the trajectories of cracks. The amount of cracks observed was from one to three. The experimental tensile strength was 4.98 MPa for CC and 4.03 MPa for SFC, so the crack width will appear when the theoretical tensile stress (σ∗) exceeds this value. 

(7)$$w=\left(\varepsilon -\frac{\Delta R^{*}f}{r^{*}}\right)\frac{2\pi r^{*}}{n}$$

From the results shown in Table 10, it is worth noting that the use of different current intensities has a direct influence on the day at which the first crack appears. The crack initiation occurs at five days for 50 μA/cm^2^ and eight days for 100 μA/cm^2^ in each concrete, though with a different tensile (σ∗) due to the microstructure of the network.

As Figure 13 shows, the pore refinement and current intensity might have a significant influence in the crack width obtained as a function of the electric charge applied. From the beginning of the test to around 1000 C, the factor with the greater influence was the current density. The lowest current density applied, 50 μA/cm^2^, caused the smallest crack widths in both concrete types, with them being even smaller for SFC50 in comparison with CC50. Above 1000 C, the crack opening evolution obtained exhibited an increasing dependency on the pore refinement, with the crack widths obtained for the concrete with silica fume, SFC, being significantly lower in comparison with the plain concrete, CC. Additionally, it is worth noting that there is a value of the electric charge, around 1200 C for CC and 1800 for SFC, from which the use of the lowest current density caused a higher crack width in comparison with the specimens subjected to the highest current density, 100 μA/cm^2^. 

## 6. Conclusions

This paper has studied the influence of the porous microstructure and current density in the initiation stage of rebar corrosion process and the crack formation caused by chlorides in reinforced concrete.

It is shown that the increase in the current density has produced a more rapid development of the corrosion process. Furthermore, it should be noted that the smaller average pore diameter size of the porous microstructure causes a lower capability to accommodate the corrosion products of the concrete, leading to greater circumferential deformations than other cases from very early ages.

The combination of an SEM analysis and strain gauges has shown a good performance when studying corrosion development at early ages. The use of wavelength dispersive X-ray spectroscopy enables data acquisition of the rust products evolution. The experimental work showed the formation of a rust layer at the interface steel/concrete at the beginning of the corrosion process, which later penetrated through the porous microstructure of the concrete around the steel bar. From the experimental results, an approximately linear relation of steel corrosion product thickness with time is deducted. 

An analytical model based on the thick-walled cylinder approach was used to reproduce the experimental results. A good level of agreement was found among the SEM observations, EDS analysis and the strain gauge measurements when a reduction factor f was included in the model for the prediction of the crack width. On the basis of calculations, it could be said that a pore refinement might influence the decrease of the reduction factor, consistent with a lesser capability to accommodate corrosion product.

The results obtained show that embedding the strain gauges in the concrete, placed as closely as possible to the steel rebar, might allow more detailed information of the circumferential strain in the concrete at local level to be available than the use of strain gauges on the concrete surface. This procedure shows improvement in the cracking models. Therefore, the crack initiation time could be estimated through conversion of the retrieved strain data into stresses, when these last ones reach the tensile strength.

In addition, from the results obtained, it could be said that the pore refinement and current intensity might have a significant influence on the crack width obtained as a function of the electric charge applied. At very early ages, up to approximately 750 C, the lowest current density applied caused the smallest crack widths in each concrete type. Later, the crack opening evolution obtained exhibited an increasing dependency on the pore refinement, with the crack widths obtained for the concrete with silica fume in comparison with the concrete without additions being significantly lower. Lastly, a value of the electric charge of around 1200 C for the plain concrete and 1800 C for the concrete with silica fume is shown, from which use of the lowest current density causes a higher crack width in comparison with the specimens subjected to the highest current density.

## Figures and Tables

**Figure 1 materials-12-02477-f001:**
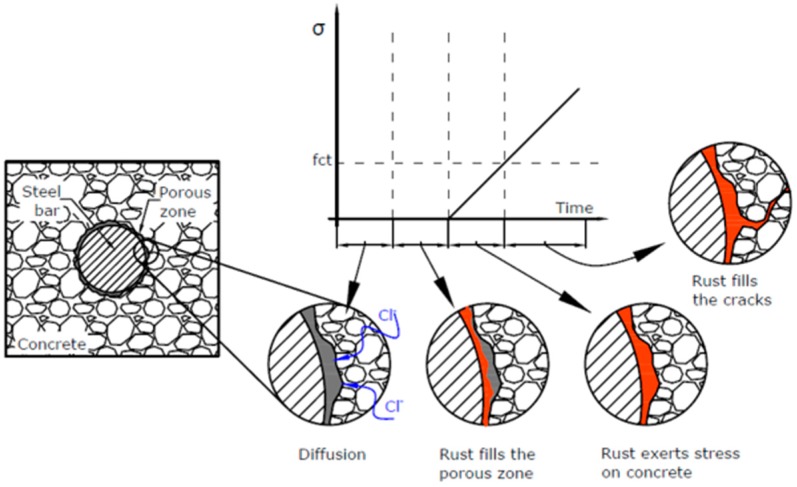
Corrosion cracking process: The three-stage-model.

**Figure 2 materials-12-02477-f002:**
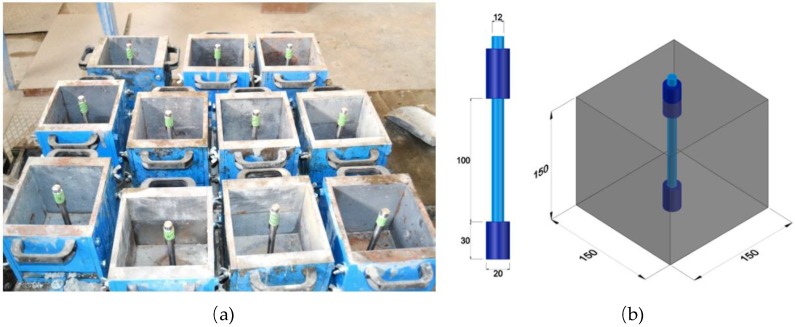
Bars in the moulds before casting: (**a**) photo of the moulds, (**b**) sketch of the specimen with the bar. (Dimensions in mm). Adapted from [14], with permission from © 2018 Elsevier.

**Figure 3 materials-12-02477-f003:**
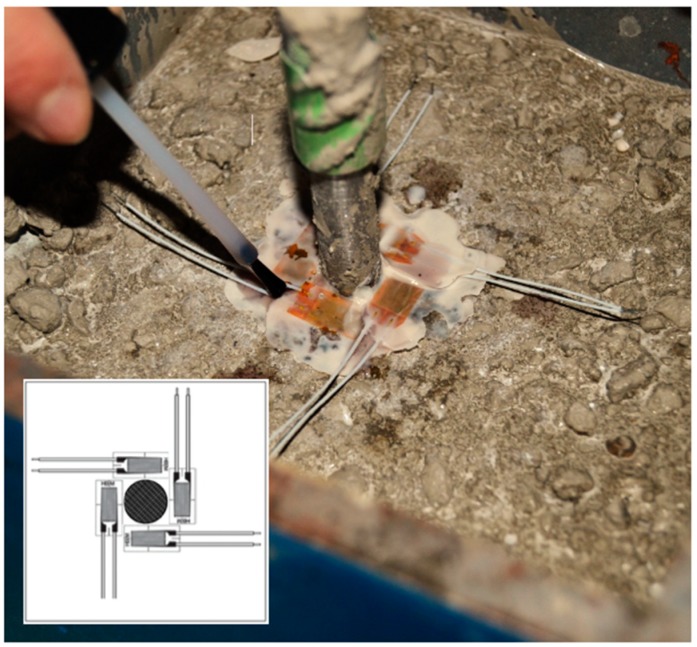
Bonding of the strain gauges before the accelerated corrosion testing.

**Figure 4 materials-12-02477-f004:**
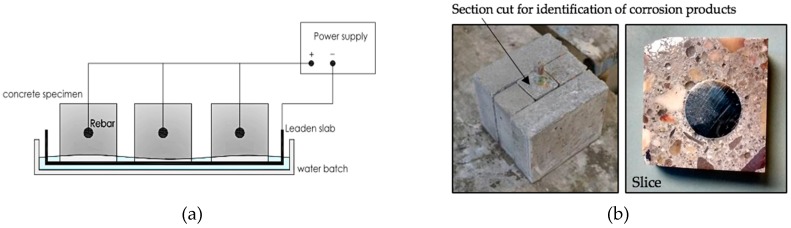
(**a**) Scheme of accelerated corrosion test. (**b**) Sketch of the cutting procedure of the specimen for SEM observation.

**Figure 5 materials-12-02477-f005:**
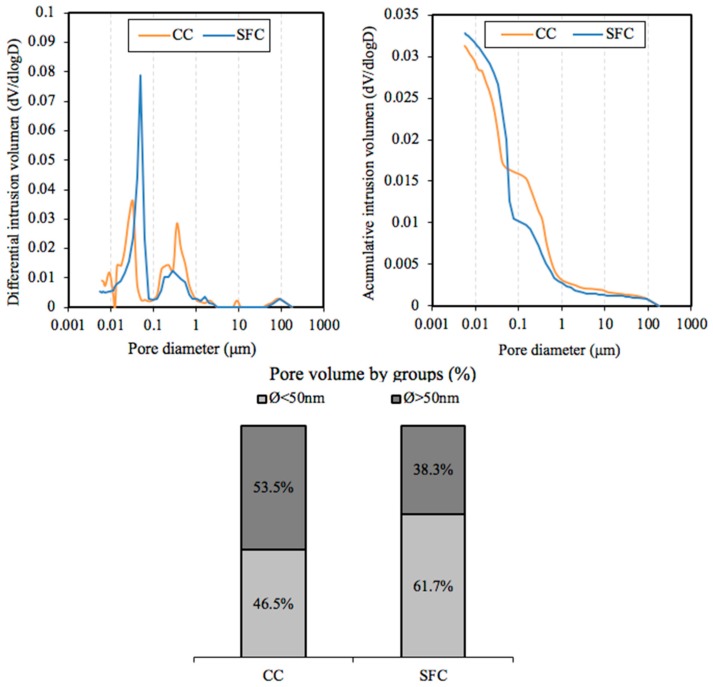
Pore size distribution for each concrete (SFC, silica fume concrete and CC, conventional concrete).

**Figure 6 materials-12-02477-f006:**
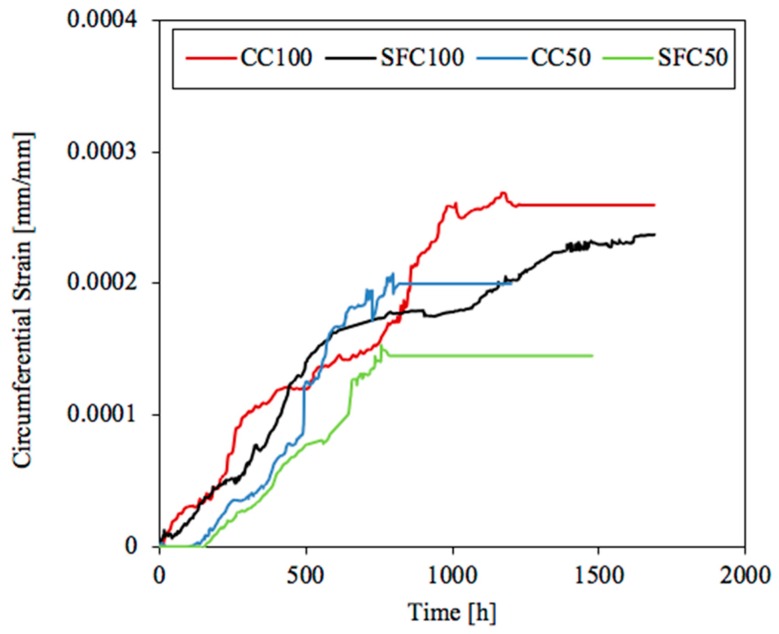
Circumferential strain around the rebar versus time for each concrete type and current density.

**Figure 7 materials-12-02477-f007:**
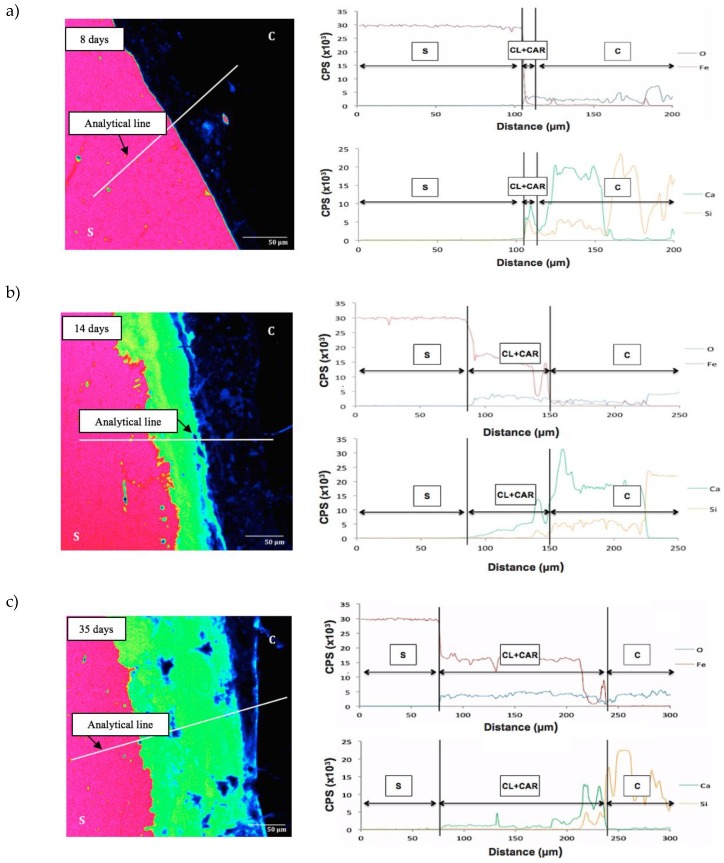
BSE image at the steel/concrete interface (C, concrete; CL + CAR, Mill scale; S, Steel) and analytical lines across the interface of CC at (**a**) 8 days, (**b**) 14 days and (**c**) 35 days with a density current of 50 μA/cm^2^. Upper curves of each age: Fe and O; bottom curves of each age: Si and Ca. Adapted from [14], with permission from © 2018 Elsevier

**Figure 8 materials-12-02477-f008:**
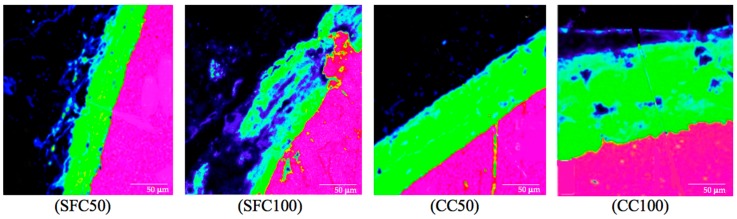
BSE image at the steel (purple)/concrete (black) interface at 14 days for each concrete (SFC, silica fume concrete and CC, conventional concrete) at at 100 µA/cm^2^ and 50 µA/cm^2^.

**Figure 9 materials-12-02477-f009:**
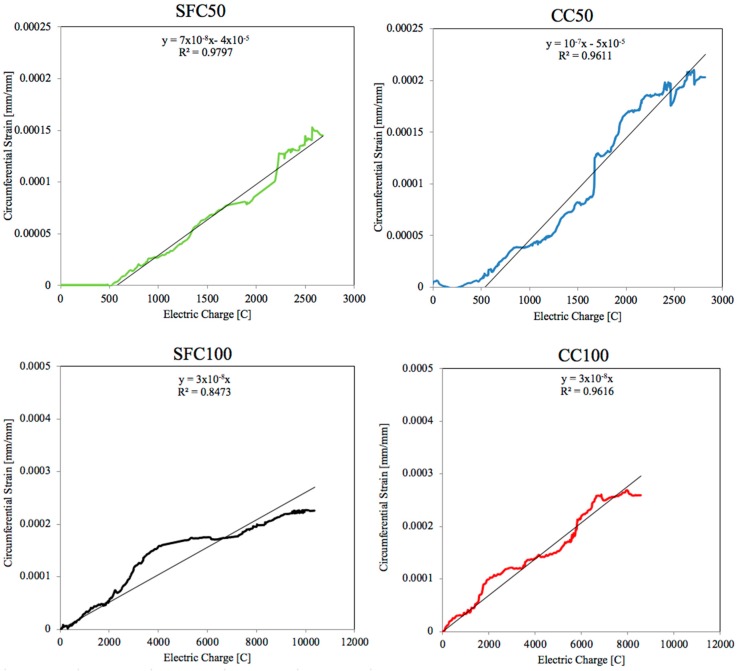
Evolution of the circumferential strain as a function of the electric charge for each specimen at 50 and 100 μA/cm^2^.

**Figure 10 materials-12-02477-f010:**
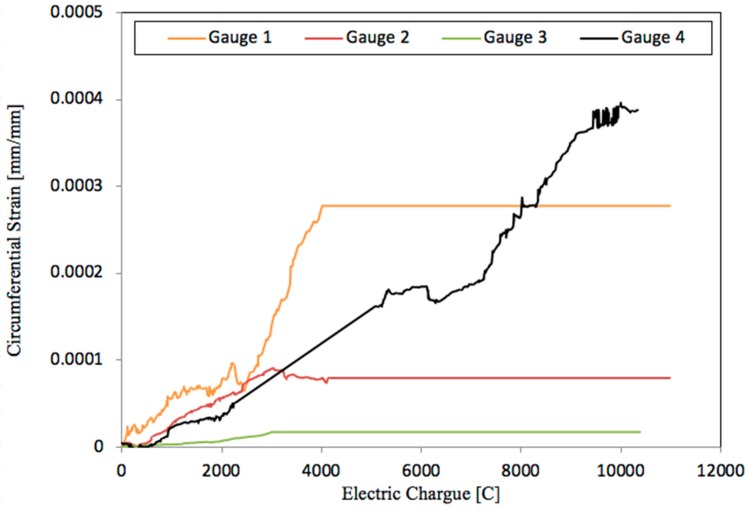
Evolution of the circumferential strain as a function of the electric charge measured by the four gauges for SFC 100.

**Figure 11 materials-12-02477-f011:**
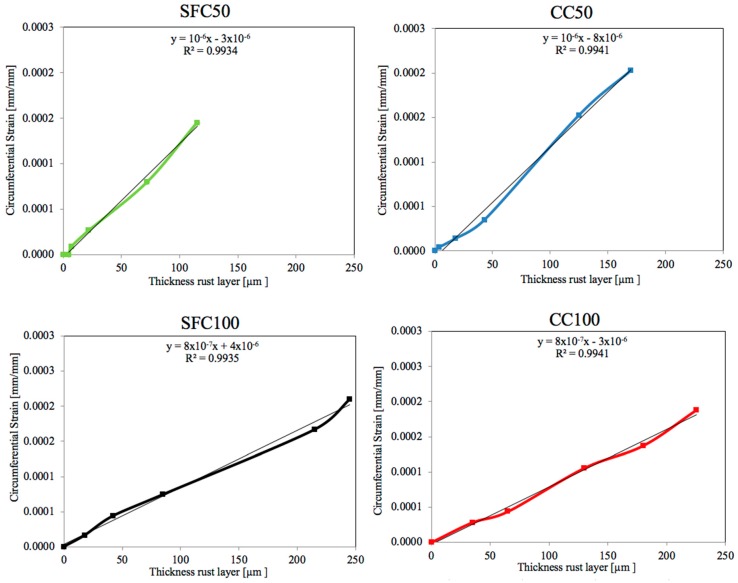
Evolution of the circumferential strain as a function of the rust layer thickness for both concretes at 50 and 100 μA/cm^2^.

**Figure 12 materials-12-02477-f012:**
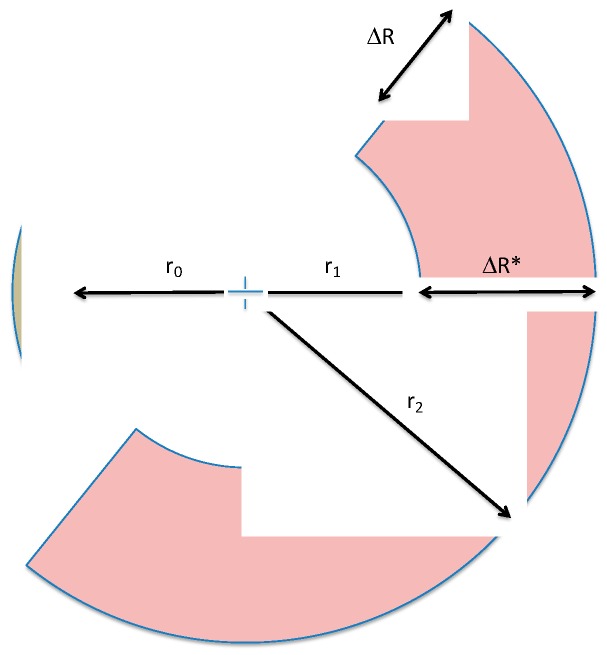
Geometrical considerations of the corrosion process in the reinforcement bar. Redrawn based on [14], with permission from © 2018 Elsevier.

**Figure 13 materials-12-02477-f013:**
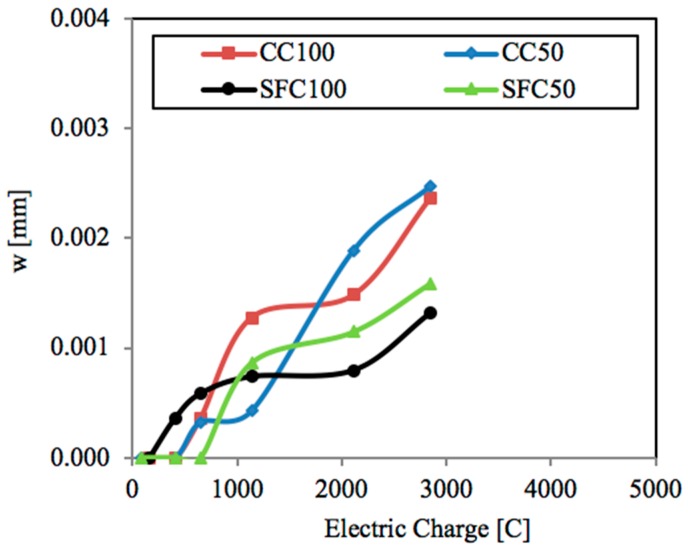
Evolution of the crack width as a function of the electric charge for each specimen at 50 and 100 μA/cm^2^.

**Table 1 materials-12-02477-t001:** Mix proportioning.

Mix	Water/Binder Ratio	Binder (kg/m^3^)		Aggregate (kg/m^3^)			Admixture (% Binder Weight)	
		Cement	Silica fume	Gravel	Grit	Sand	Superplasticizer	CaCl_2_
CC	0.45	350	-	552.2	225.4	907.8	0.9	3
SFC	0.45	280	35 *	552.2	225.4	907.8	1	3

* Efficiency factor of silica fume is taken 2.

**Table 2 materials-12-02477-t002:** Strain gauge parameters.

Type HBM	Operating Temperature Range Compensated (°C)	Resistance (Ω)	Gauge Factor	Dimensions (mm)	
K-CLY4-0100-1-120-O	−10/+45	120 ± 0.35%	2.07 ± 1.0%	Measuring grid	Measuring grid carrier
				10 × 5	8 × 18

**Table 3 materials-12-02477-t003:** Mechanical properties.

	Compressive Strength f_c_ (MPa)	Elasticity Modulus E (GPa)	Tensile Strength f_ct_ (MPa)	Fracture Energy G (N/m)
SFC	62.38	31.89	4.03	179.6
CC	65.91	33.97	4.98	172.7

**Table 4 materials-12-02477-t004:** Surface rebars electricity density and total electricity current in the specimens.

Number of Specimens	Surface Rebar (cm^2^)	Current Density (μA/cm^2^)	Total Current/Circuit (mA)
8	37.7	50	1.9
		100	3.8

**Table 5 materials-12-02477-t005:** Boundary conditions of wavelength dispersive X-ray spectroscopy (WDS, (JEOL Superprobe JXA-8900 M, Tokyo, Japan)).

Days	Element Map			Analytical Line		
	Size (Pixels)	Size Pixel (µm)	Dual Time (ms)	Accelerating Voltage (kV)	Length (µm)	Interval (µm)
1–14	400 × 400	0.5	15	20	200–250	15
26	600 × 600	0.5	15	20	250–300	15
35	600 × 600	0.5	15	20	300–600	15

**Table 6 materials-12-02477-t006:** Total porosity and effective capillary porosity for each concrete (SFC, silica fume concrete and CC, conventional concrete).

Parameter	CC	SFC
Porosity (%)	7.39	7.55
Average pore diameter (µm)	0.036	0.029
Median pore diameter (µm)	0.12	0.06

**Table 7 materials-12-02477-t007:** Thickness of the mill scale as a function of the time, for the specimens of SFC and CC with a density current of 50 and 100 μA/cm^2^.

Time (days)	Δ*R* (µm)			
	SFC50	SFC100	CC50	CC100
1	0	0	0	0
2	0	0	0	20
5	5	18	4	35
8	7	42	18	65
14	22	85	43	130
26	72	215	125	180
35	115	255	170	225

**Table 8 materials-12-02477-t008:** Relation between the calculated circumferential stress σ∗ and the $\Delta R^{*},\;$ radius in the conventional concrete.

Days	CC50			CC100		
	σ∗ (MPa)	CL + CAR (μm)	*f* (*10^−2^)	σ∗ (MPa)	CL + CAR (μm)	*f* (*10^−2^)
2	0.00	0	0	0.00	20	0
5	0.12	4	0.2	0.00	35	0
8	0.54	18	0.6	0.80	65	0.5
14	1.24	43	0.6	1.19	130	0.4
26	4.61	125	0.6	2.85	180	0.5
35	5.48	170	0.7	3.85	225	0.5
Average			0.5			0.5

**Table 9 materials-12-02477-t009:** Relation between the calculated circumferential stress (σ∗) and the $\Delta R^{*},\;$ radius in the concrete with silica fume.

Days	SFC50			SFC100		
	σ∗ (MPa)	CL+CAR (μm)	*f* (*10^−2^)	σ∗ (MPa)	CL+CAR (μm)	*f* (*10^−2^)
2	0.00	0	0	0.00	0	0
5	0.00	5	0	0.67	18	0.8
8	0.25	7	0.6	1.18	42	0.6
14	0.90	22	0.6	2.01	85	0.5
26	2.58	72	0.6	4.46	215	0.5
35	3.91	115	0.7	4.81	255	0.4
Average			0.6			0.6

**Table 10 materials-12-02477-t010:** Cracks widths of the concretes according to Equation (7) (see Figure A1 for observing the number of cracks).

Age (Days)	CC50 (f = 0.114)			CC100 (f = 0.119)			SFC50 (f = 0.087)			SFC100 (f = 0.109)		
	n	σ∗ (MPa)	*w* (μm)	*n*	σ∗ (MPa)	*w* (μm)	*n*	σ∗ (MPa)	*w* (μm)	*n*	σ∗ (MPa)	*w* (μm)
02	0	0.00	-	0	0.00	0.010	-	0.00	-	0	0.00	-
05	0	0.12	-	1	0.80	0.016	-	0.00	-	1	0.67	-
08	1	0.54	-	2	1.19	0.054	-	0.25	-	1	1.18	-
14	1	1.24	-	2	2.85	0.144	-	0.90	-	2	2.01	-
26	3	4.78	-	2	4.98	0.174	1.48	2.58	-	2	4.46	0.8
35	3	5.48	2.47	3	5.51	0.310	2.36	4.04	1.59	3	4.81	1.32

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
